# Concurrent Validation of MI-CAT(V), a Clinical Metrology Instrument for Veterinarians Assessing Osteoarthritis Pain in Cats, through Testing for Firocoxib Analgesic Efficacy in a Prospective, Randomized, Controlled, and Blinded Study

**DOI:** 10.3390/ani14050711

**Published:** 2024-02-24

**Authors:** Aliénor Delsart, Colombe Otis, Vivian S. Y. Leung, Émilie Labelle, Maxim Moreau, Marilyn Frezier, Marlene Drag, Johanne Martel-Pelletier, Jean-Pierre Pelletier, Eric Troncy

**Affiliations:** 1Groupe de Recherche en Pharmacologie Animale du Québec (GREPAQ), Université de Montréal, St-Hyacinthe, QC J2S 2M2, Canada; alienor.delsart@umontreal.ca (A.D.); colombe.otis@umontreal.ca (C.O.); vleung01@alumni.uoguelph.ca (V.S.Y.L.); emilie.labelle@umontreal.ca (É.L.); m.moreau@umontreal.ca (M.M.); marilyn.frezier@umontreal.ca (M.F.); 2Osteoarthritis Research Unit, University of Montreal Hospital Research Center (CRCHUM), Montreal, QC H2X 0A9, Canada; jm@martelpelletier.ca (J.M.-P.); dr@jppelletier.ca (J.-P.P.); 3Boehringer Ingelheim Animal Health, Fulton, MO 65251, USA; marlene.drag.ext@boehringer-ingelheim.com

**Keywords:** feline, osteoarthritis, firocoxib, pain scale, metrology, actimetry

## Abstract

**Simple Summary:**

Feline osteoarthritis (OA) diagnosis depends greatly on owner perception and experience of the disease. Owners frequently consider the emergence of OA (delayed and severe) signs as “normal” for an aged cat. Being an incurable, progressive, degenerative process, OA will require life-long treatment. There is a need for a rapid, reliable, and inexpensive diagnostic tool that reveals feline OA pain. A refined scale, the Montreal Instrument for Cat Arthritis Testing, for Use by Veterinarians (MI-CAT(V)) presented remarkable metrological properties (specific, sensitive, and reliable) over its development and validation process. In particular, MI-CAT(V) was responsive to a treatment with firocoxib, a non-steroidal anti-inflammatory drug (NSAID), and discriminated four degrees of OA pain functional severity. Firocoxib presented a clear treatment effect based on the MI-CAT(V) and on other functional objective assessments, including possible dose-response. The NSAID was safe over the three-week daily administration tested period. The cluster repartition offers new perspective for an individualized treatment care.

**Abstract:**

Veterinarians face the lack of a rapid, reliable, inexpensive, and treatment-sensitive metrological instrument reflecting feline osteoarthritis (OA) pain. The Montreal Instrument for Cat Arthritis Testing, for Use by Veterinarians (MI-CAT(V)) has been refined in 4 sub-sections, and we proposed its concurrent validation. Cats naturally affected by OA (*n* = 32) were randomly distributed into 4 groups of firocoxib analgesic (Gr. A: 0.40; B: 0.25; C: 0.15, and P: 0.00 mg/kg bodyweight). They were assessed during Baseline, Treatment, and Recovery periods using MI-CAT(V) and objective outcomes (effort path, stairs assay compliance, and actimetry). The MI-CAT(V) total score correlated to the effort path and actimetry (Rho_S_ = −0.501 to −0.453; *p* < 0.001), also being sensitive to treatment responsiveness. The pooled treatment group improved its total, gait, and body posture scores during Treatment compared to the Baseline, Recovery, and placebo group (*p* < 0.05). The MI-CAT(V) suggested a dose-(especially for Gr. B) and cluster-response. Cats in the moderate and severe MI-CAT(V) clusters responded to firocoxib with a remaining analgesic effect, while the mild cluster seemed less responsive and experienced a negative rebound effect. The MI-CAT(V) was validated for its OA pain severity discriminatory abilities and sensitivity to firocoxib treatment, providing a new perspective for individualized care.

## 1. Introduction

Feline osteoarthritis (OA) is a common cause of long-standing pain and physical dysfunction, but it remains largely underdiagnosed and undertreated [1]. The lack of a validated clinical metrology instrument (CMI), used by owners or veterinarians, explains, in part, the poor detection and consequent poor management of feline OA [2]. To be pertinent, the CMI should be valid, reliable, and repeatable [3]. In addition to these metrological properties, in practice, cost-effectiveness and feasibility must also be considered. Four types of validity must be considered: face, content, construct, and criterion validity. The face and content validity ensures that all items included in the scale cover the domain of interest (e.g., feline OA pain) and are well-weighted. While face validity is assessed by a naïve population (e.g., 1st year veterinary students), content is judged by experts (internal and external origin). Both represent the initial development of the scale. The construct validity refers to specific items of the scale that better reflect the domain of interest. It is often evaluated using correlations between the new scale and other measures of the same dimension. The criterion validity is composed of concurrent or predictive validation. It corresponds to an association, at the same or later time, between the scale and other valid assessments sensitive to a treatment (e.g., functional assessments [4,5,6]). It refers to the sensitivity and specificity of the measure [7]. The reliability informs on the error of measurement of one trial between evaluators whereas the repeatability refers to one evaluator between trials [8].

Since 2012, the Montreal Instrument for Cat Arthritis Testing, for Use by Veterinarians (MI-CAT(V)) has been developed, refined and its validation continuously published. Twelve years ago, owners and veterinarians were contacted to collect information on feline OA diagnosis, signs at home, and treatment responsiveness. It resulted in 13 items including the following: gait change, reduction in jumping and stairs use, reduction to play, decreased activity level, and coat and claw changes (i.e., content validation) [9]. After refinement (i.e., face and content validation), 8 items were included in the scale: interaction, exploration, posture, gait, body condition score, coat and claws (condition), (joint) palpation–findings, and palpation–cat reaction [10]. A pilot and a main study were performed with a cat colony without (normal cats) and with OA (confirmed by radiography). Gait, body posture, exploratory behavior, and interactive behavior demonstrated inter-item correlations, and gait detected OA. In podobarometric gait analysis measuring ground reaction forces, the peak vertical force (PVF) correlated negatively with gait and body posture, but no correlations were found for the paw withdrawal threshold (PWT) reflecting the central sensitization (i.e., construct validation and reliability). Therefore, the MI-CAT(V) appeared not to detect central sensitization, but to reflect functional impairments associated with OA pain. The scale was revised to improve its sensitivity and therapeutic responsiveness, resulting in 10 (Phase I), then 8 (Phase II), and finally 5 (Phase III) sub-sections decomposed into 25 items [11]. The body posture, gait, willingness and ease of horizontal movements, jumping, and global distance examination sub-sections were included. The scale demonstrated specific detection of naturally occurring OA in cats, with a good to excellent level of reliability, the effect of tramadol treatment (tested in Phase II and III) was notable for the jumping sub-section (i.e., construct and criterion validation and reliability) [11].

We revised the scale and proposed a simplified 4 sub-sections procedure (body posture, gait, obstacles, and global distance examination) composed of 16 items. Its responsiveness to a coxib non-steroidal anti-inflammatory drug (NSAID) treatment, firocoxib, was tested at the same time of objective functional assessments (i.e., the effort path, stairs assay compliance, and actimetry) previously validated for their sensitivity and specificity of response to treatment [4,5,6,12,13] (i.e., construct and concurrent validation). As reported, PVF and actimetry distinguished between OA and normal cats [4,5,12] and were sensitive to treatment effect on OA pain: either meloxicam PO SID administration [5], or oral transmucosal spray SID administration [14]; gabapentin PO TID administration for actimetry [11]; tramadol PO BID administration either alone [12] or associated with meloxicam but only for PVF [14]; or frunevetmab [internal data, 2021] for actimetry. Further, as recently described [6], the Stairs and Path were sensitive to firocoxib treatment in the same study design.

The main objective of this project was to validate the MI-CAT(V) discriminatory abilities and its sensitivity to firocoxib treatment. We hypothesized that -1-MI-CAT(V) measures at baseline would be correlated to previously validated functional assessments; -2-MI-CAT(V) would be sensitive to firocoxib treatment; and -3-MI-CAT(V) would allow the distinction of clusters depending on the OA pain functional severity.

## 2. Materials and Methods

### 2.1. Ethics Statement

Institutional Animal Care and Use Committees approved the protocol (#A176-BIA19F and #CEUA-Rech-1832) which was conducted in accordance with principles outlined in the current Guide to the Care and Use of Experimental Animals published by the Canadian Council on Animal Care and the Guide for the Care and Use of Laboratory Animals published by the US National Institutes of Health.

### 2.2. Animals

Adult neutered [5.5–12.5 y] cats (*n* = 32; *n* = 16 females) with an average weight of 4.8 [3.4–6.8] kg were included in the study. Cats were group-housed in lighting-, temperature-, and humidity-controlled rooms containing environmental enrichment (access to windows, perches, covered and uncovered beds, scratching posts and toys). Litters with privacy (but visible for camera assessment) were cleaned every day. A supply of fresh tap water was available ad libitum throughout the housing period. Cats were fed twice daily (morning and afternoon) with a standard, commercial diet (Purina Pro Plan^®^ Veterinary Diet^®^ Feline OM^®^), according to the manufacturer’s recommendations. Cats were weighed (kg) before each data acquisition session.

### 2.3. Selection Criteria

Cats were selected based on radiographic evidence of naturally occurring OA, screening being performed for the thoracic (carpus, elbow, shoulder) and pelvic (tarsus, stifle, hip) appendicular joints, with all requested views to confirm the radiographic OA diagnosis. A Diplomate of the American College of Veterinary Surgeons (B.LU.) reviewed and scored all X-rays independently and blindly. The radiographic score corresponds to the summation of the severity scale (0–5, null to severe OA) of the twelve joints evaluated added to the number of joint(s) affected by radiographic OA. To be selected, a cat had to present some radiographic alterations (i.e., presence of osteophytes and/or subchondral sclerosis or cyst) in at least one appendicular joint to be considered as OA. Lesions such as meniscal mineralization or enthesiophytes had to be associated with osteophytes and/or subchondral alteration to be clinically significant.

Prior to treatment, cats had no clinically significant abnormalities on complete blood count (CBC), serum chemistry, urinalysis (if applicable), behavior (not interfering with performance of required procedures), nor changes on general, neurologic, and orthopedic physical examinations other than those compatible with OA. Four weeks before the beginning of the experiment, cats were free of any treatment presenting a potential analgesic property (NSAID, tetracycline, or corticosteroids) including natural health products or veterinary diets purported to relieve or ease the clinical signs of OA.

### 2.4. Experimental Protocol

#### 2.4.1. Study Timeline

After acclimation over five weeks (W), cats were evaluated before beginning treatment for Baseline (W-3, W-2, W-1) records, then during the three full weeks of Treatment (W0, W1, W2) and through Recovery period (W3, W4, W5). The number of assessments during each period for each outcome is given in the next section. All evaluators were blinded to the treatment status.

#### 2.4.2. Treatment

Osteoarthritic cats were randomized into four groups to receive one of the following firocoxib treatment doses, for a 21-day period by the oral route (Gr. A: 0.40, Gr. B: 0.25, Gr. C: 0.15 and P Gr.: 0.00 mg/kg bodyweight SID). The treatment group refers to the pooled Gr. A, B, and C, whereas the placebo group refers to the P Gr. alone. Cats were continuously assessed by two registered veterinary technicians under the supervision of one registered doctor in veterinary medicine. In addition to these daily general health observations, a complete physical examination conducted by an independent licensed veterinarian, CBC and serum biochemistry analyses, as well as urinalysis (cystocentesis under reversible dexmedetomidine-hydromorphone sedation) were conducted during acclimation (W-7), Treatment (W2), and Recovery (W5).

### 2.5. Outcome Measures

#### 2.5.1. Highly Refined Pain Assessment Method—Montreal Instrument for Cat Arthritis Testing for Use by Veterinarians (MI-CAT(V))

The MI-CAT(V) is a feline OA pain scale, see Appendix A in the Appendix A. The pain scale is composed of 4 sub-sections as follows: (1) Body posture; (2) Gait; (3) Obstacles; and (4) Global distance assessments. These four scores are added for a maximum of 60, and the MI-CAT(V) total score is expressed in percentage of alteration. The jump height of the obstacle sub-section was reported. Evaluations were conducted by a veterinary medicine doctor (E.TR.), who remained blinded to the treatment period and group, at three distinct timepoints: Baseline (W-1), Treatment (W2), and Recovery (W5).

#### 2.5.2. Stairs Assay Compliance

The stairs assay compliance (Stairs) measured the functional capacity of the cat to complete a series of going up and down a 16-stairs (ceramic tiles, height of 20 cm, width of 117 cm, and depth of 28 cm) assay, previously validated in OA cats [6]. After five weeks of acclimation using positive reinforcement, cats were encouraged to do the maximum number of up and down passages during a four-minute period. The stairs assay outcome was conducted at the following timepoints: Baseline (W-3, W-1), Treatment (W2), and Recovery (W5). During both baseline assessments, the median value of completed up and down passages for the population sample (*n* = 31; one evident outlier) was calculated to be 7 and named the “finish-line”. The number of cats (expressed in percentage) in each group crossing this “finish-line” was assessed at each subsequent timepoint.

#### 2.5.3. Effort Path

The innovative effort path (Path) was validated to assess functional impairments in OA cats [6]. Cats were trained during five weeks to cross the path at a comfortable speed using positive reinforcement. Three to five trials were obtained for each cat and were submitted to a podobarometric gait analysis at a resolution of 1.4 sensels/cm^2^. Measurement included (1) velocity of the cat on the platform; (2) PVF as cats jumped down and up from the calibrated pressure-sensitive mattress (Matscan^®^ System, Tekscan Inc, Boston, MA, USA); and (3) the number of frames (reflecting the time to passing the pressure-sensitive mattress). The Path was conducted at Baseline (W-3, W-1), Treatment (W2), and Recovery (W5) periods.

#### 2.5.4. Telemetered Actimetry (Locomotor Activity Monitoring)

The motor activity was assessed using a collar-attached accelerometer-based activity sensor (Actical, Mini-Mitter/Respironics, distributed by Bio-Lynx Scientific Equipment Inc., Murraysville, PA, USA) maintained in place from W6 to W5 after acclimation of the cats to wearing the device. The device was set for local time and configured to record an epoch event of 1 count per minute. The amplitude of each count was subsequently translated to a numeric value (from 0 to infinity) referring to the intensity of actimetry intensity count. To exclude periods where human activity and handling interfered with the cats’ activity, only 4 days per week (Friday, Saturday, Sunday, and Monday) and night-time between 5:00 P.M. to 6:59 A.M. (starting for Friday, Saturday, and Sunday) were considered for the analyses and later referred to as weekend (total of 12 timepoints). Baseline initial condition recording was for the first six weekends, with the next three weekends as Treatment (W0, W1, and W2) and the last three ones as the Recovery period (W3, W4, and W5).

Time sequencing was supported by data from our previous studies, as well as published data [5,10,14]. Night-time actimetry monitoring (NAM) recordings were considered for the analyses of data recording during night-time between 5:00 P.M. to 6:59 A.M. For each weekend, there were three periods of night-time recording. The duration of each period was 14 h (total 42 h per weekend), and data were either the sum of the actimetry intensity counts recorded over this 3 × 14 h period or the sum per hour. Data were expressed as the group-average total intensity counts over the weekend (42 h), or over each period of NAM (14 h).

### 2.6. Statistical Analysis

The intraclass coefficient of correlation (ICC), based on a single measurement, absolute agreement, and two-way mixed-effects model, was used to determine the reliability between NAM Baselines and interpreted as follows: >0.81 very good, >0.61 good, >0.41 moderate, >0.21 fair, and <0.20 poor reliability [15]. Two cats were excluded as facing medical issues not related to tested investigational veterinary product (one in P Gr., one in Gr. A), and the analysis was performed with *n* = 30 cats.

For the construct validation, the correlations between the MI-CAT(V) scores and the functional assessments were calculated using the Spearman rank’s test. The interpretation of the coefficient of correlation (Rho_S_, ρ) was as follows: 0–0.35 = weak, 0.36–0.70 = moderate, and 0.71–1.00 = strong agreement. For this test, the Baselines were included as independent timepoints.

The normality was calculated using the Shapiro–Wilk test. The treatment effect evaluated by MI-CAT(V) was assessed using a linear mixed model with the interaction of time (i.e., timepoints) and treatment groups as fixed factors. The time was included as a repeated factor, and the α threshold was adjusted for multiple comparisons using the Bonferroni correction. For actimetry, inferential analysis was performed without adjustment using a generalized linear mixed model with Baseline as covariate, subject and timepoints as repeated effects, the hour as random effect, and the interaction of time and treatment groups as fixed factors. By distinguishing cats in clusters according to their MI-CAT(V) scores, the responsiveness to treatment was analyzed using Fisher test comparisons and the Mann–Whitney non-parametric test; for the inter-group and within-time analysis, pairwise comparisons were performed. The percentage of responder cats was calculated for MI-CAT(V) and NAM as follows: a decrease in MI-CAT(V) over 15% vs. Baseline; and a positive or null slope after NAM linear representation (from W-5 to W2 for the treated cats or W-5 to W5 for the P Gr.). Analyses were performed using Fisher test comparisons. The significance level was set as *p* < 0.05 and *p* < 0.1 as a tendance. Analysis was performed using Sigma Plot (v.12.0) and SPSS (v.26) statistical software.

## 3. Results

Over the three-week period of daily administration, firocoxib was very well accepted. Not only was no serious adverse event detected, but also for the physical examination from W-7 to W5, no pathological condition which could be related to any treatment was detected. At Treatment (W2), two cats presented serum biochemical analysis with a slight increase in renal outcomes, but both cats were in the P Gr. In the daily observations, there were reports of gastrointestinal disturbances (soft feces, vomiting, regurgitation), but they were all transient, and in the ten cases reported, five were from the P Gr. OA cats.

### 3.1. Construct Validation: MI-CAT(V) Is Correlated to Validated Objective Functional Assessments

The reliability between NAM Baseline acquisition sessions was good, at 0.68 (95% confidence interval: 0.43–0.84; *p* < 0.001). At Baseline, there was a moderate negative correlation between MI-CAT(V) total score and Path—Velocity, Path—Thoracic limb PVF and NAM (Table 1; Rho_S_ = −0.303 (*p* = 0.019) to −0.501 (*p* < 0.001)). There were also moderate negative correlations between gait, obstacles, and global distance examination sub-sections with Path—Thoracic limb PVF and NAM (Rho_S_ = −0.373 (*p* = 0.004) to −0.524 (*p* < 0.001)). The obstacles sub-section was moderately negatively correlated to the Path—Velocity (Rho_S_ = −0.405 (*p* = 0.001)) and moderately positively correlated to the Path—Time of completion (Rho_S_ = 0.408 (*p* = 0.002)).

Globally, MI-CAT(V) total score and most individual items were correlated to functional impairments associated with OA pain, such as podobarometric gait analysis (Path—Velocity and Thoracic limb PVF) and actimetry (NAM).

### 3.2. Criterion Validation: MI-CAT(V) Is Sensitive to a Firocoxib Treatment

#### 3.2.1. MI-CAT(V) Sensitivity of Response to Firocoxib

According to univariate analysis, the P Gr. remained stable (*p* = 0.622), whereas the MI-CAT(V) total score of the pooled treatment group varied within time (Type III *p* = 0.007). The pooled treatment group scores were lower during the Treatment period compared to both Baseline (by 28%; *p* = 0.013) and Recovery (by 26%; *p* = 0.028). During Recovery, MI-CAT(V) scores were worsened, and did not significantly differ from the Baseline (*p* = 1.000). Also, the pooled treatment group decreased its MI-CAT(V) total score compared to P Gr. during Treatment (*p* = 0.021), and this tended to remain during Recovery (*p* = 0.066) (Figure 1). The MI-CAT(V) power of effect was 83% (*p* = 0.004) for the groups comparison.

Positive responsiveness to firocoxib treatment for Stairs and Path has been previously presented [6]. For actimetry, a MI-CAT(V)-like clear treatment effect was observed for the pooled treatment group (Type III *p* < 0.001) (Figure 2) with a significant increase in NAM compared to Baseline (by 17%; *p* = 0.009) and a significant difference to P Gr. during the Treatment phase (*p* = 0.004). The latter between-groups difference remained significant during Recovery (*p* = 0.005). The P Gr. showed no significant change (*p* > 0.368) over time. The NAM power of effect was 99% (*p* < 0.001) for the groups comparison. The MI-CAT(V) was similarly sensitive to NAM to detect responsiveness to treatment (Table 2). A large majority of cats in the pooled treatment group (≥65%) responded to Treatment, whereas the P Gr. remained under 14% of responders in each tested outcome at Treatment (*p* < 0.05; Table 2).

Body posture and gait sub-sections were sensitive to pooled treatment group change (Type III *p* < 0.018), whereas P Gr. remained stable according to univariate analysis (*p* > 0.077). The body posture scores were 39% lower for the pooled treatment group at Treatment compared to Baseline (*p* = 0.011) and were lower than P Gr. during Treatment (*p* = 0.043) (Figure 3). The gait scores of the pooled treatment group evolved similarly with a significant decrease during Treatment compared to Baseline (by 46%; *p* = 0.002) and Recovery (by 44%; *p* = 0.005). Also, gait scores were lower than P Gr. during Recovery (*p* = 0.040) (Figure 3). A trend was highlighted for the global distance examination score (Type III *p* = 0.074) with a decrease for the pooled treatment group compared to P Gr. (*p* = 0.012) during Treatment. The statistical model was not able to determine significant change for the obstacles sub-section (Type III *p* = 0.150).

#### 3.2.2. Dose Response

The fixed effect time by groups interaction was close to significance for the MI-CAT(V) total score (Type III *p* = 0.051) sustained by the univariate analysis of the Gr. B (*p* = 0.003). The Gr. B MI-CAT(V) scores (see Figure 1) significantly decreased at Treatment compared to Baseline (by 43%; *p* = 0.014) and Recovery (by 41%; *p* = 0.029). Further, the univariate analysis indicated within-time change for the Gr. B body posture (*p* = 0.012), gait (*p* = 0.002), and global distance examination (*p* = 0.017) scores. A within-time change for the gait of the Gr. C was also detectable (*p* = 0.024). The fixed effect time by groups interaction was significant for the gait scores (Type III *p* = 0.007), the Gr. B scores being lower during Treatment compared to Baseline (*p* = 0.015) and to Recovery (*p* = 0.010). A trend was highlighted for the gait scores of Gr. C, they slightly decreased at Treatment compared to Baseline (*p* = 0.097) and to Recovery (*p* = 0.069). The statistical model was not able to determine any significant change for the obstacles sub-section (Type III *p* = 0.274).

### 3.3. Criterion Validation: The Clusters Influence on Responsiveness to Treatment

#### 3.3.1. Clusters Stratification Determination

According to Baseline correlations, three MI-CAT(V) clusters were determined: mild (MI-CAT(V) score ≤ 20%; *n* = 10), moderate (MI-CAT(V) score between 21–35%; *n* = 9), and severe (MI-CAT(V) score > 35%; *n* = 4) OA functional injuries. The descriptive data are presented in Appendix A of the Appendix A.

At Baseline, cats in the mild cluster were more active during NAM than cats in the moderate cluster (*p* = 0.034). Also, they trotted faster reflected by the Path—Velocity (*p* = 0.004), had higher values for Path—Thoracic limb PVF (*p* = 0.004), and tended to be more active during NAM (*p* = 0.057) than cats in the severe cluster. Cats in the moderate cluster performed faster for Path—Velocity (*p* = 0.013), and they had a higher Path—Thoracic limb PVF (*p* = 0.024) than cats in the severe cluster. The jump height of cats performing the obstacle sub-section was lower for the severe cluster compared to mild and moderate clusters (*p* = 0.05)

#### 3.3.2. Clusters and Treatment Effect

Mild OA cats were responsive at 50% in MI-CAT(V) during the Treatment period. They worsened their MI-CAT(V) score at Recovery compared to Baseline (*p* = 0.001) and to Treatment (*p* = 0.028) periods, reflected by the number of responders that dropped to zero compared to the Treatment period (*p* = 0.033) (Table 2 and Figure 4). Further, at Treatment, they had Path—Pelvic limb PVF values higher than cats in the moderate and severe clusters (*p* < 0.037), and at Recovery, they performed less than moderate cats for the Stairs—Finish Line Up (*p* = 0.023).

The moderate and severe clusters maintained their responsiveness within-time, and they improved their MI-CAT(V) total scores both during Treatment (*p* < 0.004) and Recovery (*p* < 0.049) periods compared to Baseline (Table 2 and Figure 4). Cats in the moderate cluster had an increase in their Path—Pelvic PVF values at Recovery compared to Baseline (*p* = 0.013) and Treatment (*p* = 0.030). Also, they were more performing at Stairs—Finish Line Up and Down at Recovery compared to Baseline (*p* < 0.021).

Mild OA cats exhibited worsening in their sub-section scores, with higher scores during Recovery compared to Treatment for body posture (*p* = 0.056) and compared to Baseline for gait, obstacles, and global distance examination scores (*p* < 0.029). Also, during Treatment, mild OA cats showed improvement with lower body posture scores than those with moderate OA (*p* < 0.036). The latter improved their gait and obstacles scores both during Treatment (*p* < 0.002) and Recovery (*p* < 0.043), whereas global distance examination scores were improved only at Treatment (*p* = 0.006). The severe OA cats evolved similarly; they improved all sub-sections during Treatment (*p* < 0.022), the body posture and obstacles scores remain lower at Recovery compared to Baseline (*p* < 0.015), and a trend was noted for the gait decrease (*p* < 0.072).

## 4. Discussion

Veterinarians have the task of maintaining optimal animal health for their patients. However, they are faced with a lack of rapid, reliable and inexpensive diagnostic tools in the case of chronic OA pain. Feline OA pain prevalence in the geriatric population has been reported to be 80% [16,17]. Although based on radiographic and clinical diagnosis, since the sample size was low in these studies (*n* = 100 cases), there is an increased risk of overestimation. However, the OA condition in cats presents a paradox of a high prevalence, while at the same time being largely under-diagnosed. Indeed, the diagnosis is based on the feline’s anamnesis [9], the physical examination, and the radiography which depends greatly on the owner’s budget. Even if physical examination and radiographies are performed, the lack of correlation between these tools and the painful experience leads to an under-management of OA pain [18,19,20]. The functional impairment and possible peripheral and central sensitization result in decreased animal welfare, increased behavioral problems, and negative impact on the human–cat bond [21,22]. A firocoxib-based treatment is commercialized for canine and equine OA pain (Previcox^®^ or Equioxx^®^; Boehringer Ingelheim), but no NSAID is currently approved in North America for safe and long-term control of feline OA pain. Clinically, although the caregiver placebo effect is known, the responsiveness to treatment is based on the owner’s perception of mobility (mostly gait and jumping) and activity level [23,24,25]. A recently licensed treatment against feline OA pain, namely frunevetmab, a felinized anti-nerve growth factor (NGF) monoclonal antibody presented, in the efficacy outcome measure, a success rate of 76% at day 56 (post first administration) [26]. However, it was counterbalanced by a placebo success rate of 65% (*p* = 0.03), which continued to increase up to 68%, leading to an absence of statistical difference with the treated group (76%) at day 84 (*p* = 0.08). Clearly, this indicates a major failure in the metrological properties of the Client Specific Outcome Measure CMI used in this study [26].

There is a need for a validated tool for veterinarians that is non-invasive, non-expensive, and sensitive to treatment, to allow for better management and restoration of the animal’s quality of life. The development of the MI-CAT(V) scale began 12 years ago and was refined as the studies progressed:
−The sensitivity, specificity and reliability were previously validated [10,11] and confirmed during this study for discriminatory ability and reliability.−The face, content, and construct validation excluded [10], first, all hands-on evaluations, namely, body condition score, coat and claws condition, joint palpation/manipulation (findings, as well as cat reaction) as being not sensitive, and/or not specific to the OA condition, or presenting poor reliability.−Four subsequent phases of construct validation were required to succeed to the present version. In the first phase 0 [10], no analgesic treatment was used, it involved 32 OA and 6 normal cats, and the scale included 13 sub-sections. In phase I [11], the tested analgesic was gabapentin in 7 OA and 5 normal cats. The scale included 8 sub-sections, and an exhaustive and rigorous Board-certified veterinary surgeon orthopedic exam conducted a palpation/manipulation of each axial segment and all appendicular joints. Once again, there was a failure in discriminatory ability and reliability. In phase II and III [11], the tested analgesic was tramadol, and, respectively, involved 15 OA, 5 normal cats, and 13 OA, 6 normal cats, and included 8 and 5 sub-sections, respectively.−Overall, including the present study, the validation process concerned 106 OA and 24 normal cats. The present, simplified, MI-CAT(V) version includes 4 sub-sections and 16 items to score. In its preceding version, the pain scale demonstrated its discriminatory ability and partial responsiveness to tramadol treatment, where the scale distinguished naturally occurring OA and normal cats and was sensitive to treatment for the jumping sub-section [11].

The aim of this study was to validate the new refined MI-CAT(V) scale by using concurrent validated objective assessments applied in feline OA research [4,5,6,12,13,14,25,27]. Spearman correlations were performed between functional (i.e., the Effort path, Stairs, and actimetry) assessments with the MI-CAT(V) total or sub-sections scores. The functional assessments could be divided in controlled (Path and Stairs) or unconstrained (actimetry) evaluation of the mobility. The MI-CAT(V) total score was poorly to moderately negatively correlated both for functional assessments in unconstrained (Rho_S_ = −0.453, *p* < 0.001) or controlled (Rho_S_ = −0.501, *p* < 0.001 to −0.303 *p* = 0.019)) environments at Baseline. Cats with lower MI-CAT(V) total scores (mild and moderate OA clusters) trotted faster (Path—Velocity) and had higher Path—Thoracic limb PVF than more affected clusters. This is in agreement with previous findings in podobarometric gait analysis, where PVF of OA cats was lower than PVF of normal cats, and a PVF increase was correlated to lower gait and body-posture scores [4,10,12]. Cats less affected according to MI-CAT(V) had also higher motor activity and, as demonstrated, an increasing or stable NAM was associated to lower pain (analgesic action) [5,11,12,14,28]. A moderate negative correlation was found for the obstacles sub-section with the Path—Thoracic limb PVF (Rho_S_ = −0.524, *p* < 0.001). This is of particular interest as a previous study showed that only the obstacles sub-section was able to discriminate a tramadol treatment effect [11]. Our current study determined that the MI-CAT(V) is correlated with functional impairment objective assessments, such as PVF and NAM (unconstrained mobility), that strongly suggests its discriminatory abilities. In fact, the MI-CAT(V) score reflects the functional alterations and the associated pain, at this level of investigation.

The MI-CAT(V) total score, body posture, gait, and global distance examination sub-sections were sensitive to the firocoxib treatment effect. While the placebo group remains stable within-time, the pooled treatment group improved its MI-CAT(V) total scores, body-posture, and gait scores at Treatment compared to Baseline, and a tendency for a remaining effect was observed at Recovery for the total, body posture, and gait scores, with a difference vs. P Gr. at this timepoint. It is interesting to note the clear firocoxib treatment effect detected by MI-CAT(V) and NAM, with a statistically significant change (by −28%, and +17%, respectively in the Treatment vs. the Baseline period), and a statistically significant between-groups difference at Treatment for P Gr. vs. the pooled treatment group. A clear treatment effect (significant intra- and inter-group difference) was also observed with the Stairs—Finish Line Up assessment [6], when (only) a within-time significant difference was present for Stairs—Finish Line Down, as well as all outcomes of the podobarometric gait analysis (Path—Thoracic and Pelvic limb PVF, Velocity and Time of completion). The percentage of responders in MI-CAT(V) and NAM (see Table 2) at the Treatment period was highly significant (>65%), when, at the same time, these outcomes were not responsive to the placebo (<14% response rate). At the Recovery period, the percentages of responders in the pooled treatment group remained elevated for MI-CAT(V) compared to the P Gr., and a remaining effect (statistically significant difference vs. Baseline) was present at Recovery for all Path and Stairs outcomes [6]. For MI-CAT(V) and NAM, the averaged value at Recovery came back to Baseline level (see Figure 1 and Figure 2), but the difference between the pooled treatment group and P Gr. remained present (*p* = 0.066, and *p* = 0.005, respectively). Such a remaining effect observed with firocoxib is interesting as a negative rebound effect was observed in several studies [5,29,30,31] when stopping meloxicam, another NSAID, in OA cats. First observed on actimetry monitoring in a colony of OA cats receiving the lowest dose of meloxicam (0.025 mg/kg PO SID, 4-week) and not on higher doses [5], it was subsequently confirmed on client-owned OA cats using different CMIs for a 10-week (0.05 mg/kg PO SID) [29] or a 3-week (0.035 mg/kg PO SID) [30,31] treatment period. Moreover, the validated CMI, MI-CAT(C), filled in by caretakers/owners, correlated negatively with NAM, reflecting the changes under meloxicam treatment (−18% for MI-CAT(C), +24% for NAM), as well as the negative rebound effect [31].

The statistical model was also able to discriminate a dose-response for the Gr. B MI-CAT(V), and similar trends were observed for Gr. A and C (see Figure 1), lacking some analysis power, particularly in Gr. A with a reduced sample size (*n* = 7). Gr. B was the dose that had the best improvement efficacy during Treatment compared to both Baseline and Recovery and to the placebo group for the MI-CAT(V) total score, as well as for Path—Velocity and Path—Time of completion. The changes observed on MI-CAT(V) scores are mostly attributable to the sub-sections body posture and gait, when global distance examination had less influence. Curiously, no treatment effect was detectable with the obstacles sub-section. This could be explained by the inter-individual variability and the item sensitivity: obstacles could present good discriminatory ability but a lower responsiveness to treatment (as confirmed by jump height with the severe cluster jumping from lower height (*p* = 0.05) but not responding to treatment throughout the study). Indeed, each sub-section is composed of items to evaluate both thoracic and pelvic limbs; internal consistency analyses (Cronbach α = 0.83) suggested no item should be dropped, and inter-rater (ICC [95% confidence interval] = 0.70 [0.51–0.83]) and intra-rater (0.67 [0.09–0.89]) reliabilities were good. The latter metrological analyses were not presented to not overload the manuscript. By adding the responsiveness to firocoxib treatment, the MI-CAT(V) appears to be a highly validated and performing feline OA pain scale.

Regarding the dose-response effect, the cluster stratification should be taken into consideration. Three clusters were distinguished according to the impairment severity determined by the MI-CAT(V) total score. This stratification was validated at Baseline with the functional evaluations, when mild OA cats performed better at the Path and during NAM than moderate OA cats, and the latter cluster performed itself better than severe OA cats. Although the number of cats in the severe cluster (*n* = 4) was low, this sample is relevant to the severity of OA cats previously reported [32]. Interestingly, the treatment effect varied with the MI-CAT(V) clusters, with the moderate and severe OA cats being highly responsive to firocoxib, and their effect on MI-CAT(V) remaining during Recovery. The latter was particularly present for the moderate clusters as it was observed during Recovery with a positive remaining effect on Path—Pelvic limb PVF and Stairs—Finish Line Up and Down. The situation was inverse for the mild cluster: its MI-CAT(V) deteriorated (increased) during the Recovery period when compared to Baseline, clearly indicating a negative rebound effect after Treatment withdrawal. Since a cluster-dependent response was observed, could the cluster stratification have influenced the dose-response? Indeed, the cluster repartition was not homogenous between dose-groups, mild and moderate clusters being well distributed in the four dose-groups (*n* = 2–4 cats per group), whereas the severe cluster was mostly present in the Gr. B (*n* = 3 cats) and absent in the Gr. C. This confirms the improvement of the Gr. B cats under treatment, as it was mostly composed of cats with moderate and severe OA, the most responsive clusters to firocoxib. With regards to the influence of the cluster stratification in predicting the treatment effect, this discriminatory ability is another strength of MI-CAT(V) and must be determined at Baseline to better homogenize the severity impairments into treated dose-groups, and eventually predict the response to treatment.

Distinguishing clusters to better understand the progression of OA and enable better pain management has gained momentum in recent years. Several clustering has been proposed in OA human patients. First, based on structural phenotype, three dominant clusters were described and associated with either low tissue turnover (non-progressors), structural damage (longitudinal progression), or systemic inflammation (characterized by a sustained or progressive pain) [33]. In the case of feline OA, obtaining such biomarkers could determine the correlation between structural and functional impairments assessed by MI-CAT(V) and predict the OA progression. Structural severity was recently classed in feline stifle OA [34], and severe OA was reported in 10–16% of the OA-affected joints, an occurrence like the one (4/23 = 17%) reported in the present study. The peripheral and central sensitization contributed to distinguish other clusters in chronic pain [35]. Notably, in humans, a higher sensitivity was highlighted in a high-pain but low knee OA grade cluster [19]. Also, a cluster with higher sensitivity and moderate temporal summation was predictive to persistent pain in the following 2 years [36]. Feline OA pain can also lead to centralized sensitization and facilitatory temporal summation, as assessed by quantitative sensory testing [37] and functional cerebral imaging [38]. As no MI-CAT(V) section was constructed to detect hypersensitization, a complementary assessment using quantitative sensory testing might reflect the somatosensorial evaluation. This remains an avenue to explore. Also, the quality of life of OA cats could be assessed with the VetMetrica^TM^ HRQL tool. This scale is addressed to veterinarians and sensitive to clustering, and quite intuitively, the quality of life decreased as the severity of the disease increased [32]. Determined correlation with the MI-CAT(V) could be interesting to complete the affective-motivational and cognitive-evaluative components of the feline OA pain. Furthermore, as reflected by the correlations and the responsiveness to treatment, the MI-CAT(V) and the functional assessments are complementary to evaluate the sensory-discriminatory component of OA pain. Taken together, these tools allow the assessment of the feline OA pain in its globality leading to personalized treatment, preventive or curative, targeting or not the sensitization. Currently, multimodal, integrative management of the feline OA pain [39] presents more characteristics of “one size fits all” than a personalized plan, including analgesic medications [40], biologics (regenerative) medicine, dietary modifications [41,42,43], nutraceutical supplementation [43], environmental adaptations, and complementary and alternative medicine such as physical rehabilitation or acupuncture [44], without real evidence of benefits or deleterious effects.

Even if the MI-CAT(V) scale was concurrently validated through firocoxib testing, some limitations were present: the sample size led to a possible statistical error of type II mostly during the clustering; Also, cats were assessed in their globality and were not grouped according to their affected joints leading to a possible heterogeneity in the population; however, this reflects the patient’s reality. Finally, the scale was completed by one evaluator (E.TR.) familiar with the MI-CAT(V), and only the firocoxib treatment was tested; further drugs and other evaluators need to be included to fully validate the scale sensitivity and reliability.

This study demonstrated the validity of the MI-CAT(V) for its discriminatory abilities and responsiveness to treatment, and its evaluation would present many advantages when performed during the veterinary consultation. It took, on average, 9.90 (2.15) min per cat to complete the scoring, and the edited Manual for MI-CAT(V) Use (see Appendix A in the Appendix A) is largely facilitating the task. This will allow the cat’s inclusion in a cluster and could lead to manage its pain in a more personalized manner: This new stratification will enhance the cat’s welfare and its chronic OA pain management. Further studies should investigate the effect of other pharmacological (NSAID, opioid, anti-NGF monoclonal antibody, etc.) or alternative therapies (specific diet, complementary and alternative medicine, biologics, etc.) for each cluster, and end up specifying the best indication for use of each treatment, alone or in synergy.

Such exciting results generate numerous avenues for prospecting: to question previous studies with no detectable treatment effect, which could be due to clusters repartition. One of the next validation steps will be to test the MI-CAT(V) between different evaluators in research and hospital environments. In a clinical consultation, if, despite following the recommendations to minimize stress, the cat is still reluctant to move, a video analysis taken by the owner at home should be assessed [45].

## 5. Conclusions

The main points to be drawn from this study are as follows: (1) the MI-CAT(V) is a specific, sensitive, reliable, and valid tool to discriminate the functional severity of OA in cats. It could be used to stratify the OA level of impairment with normal cats (<9%) [33], mild OA (9–20%), moderate OA (21–35%) and severe OA (>35%) degrees; (2) firocoxib was safe over the three-week period of daily administration, and presented a clear treatment effect on MI-CAT(V) and on other functional objective assessments, including possible dose-response; and (3) the cluster repartition offers a new perspective for individualized treatment care. Based on MI-CAT(V) stratification, the mild OA cluster seemed less responsive and experienced a negative rebound effect at firocoxib withdrawal, but cats in the moderate and severe OA clusters responded more to firocoxib with a remaining analgesic effect during Recovery.

## Figures and Tables

**Figure 1 animals-14-00711-f001:**
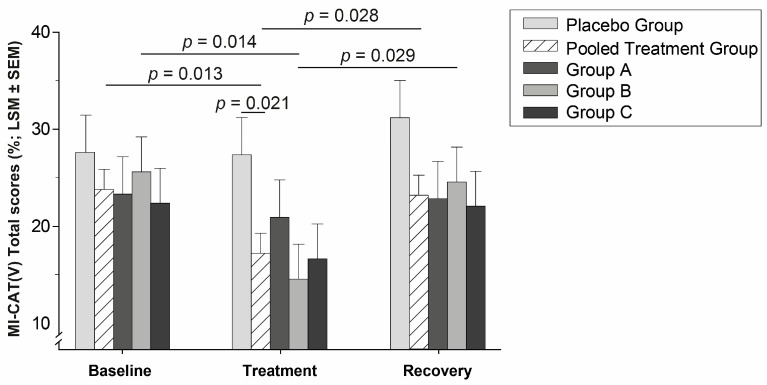
MI-CAT(V) total score evolution of the different groups during Baseline, Treatment, and Recovery periods.

**Figure 2 animals-14-00711-f002:**
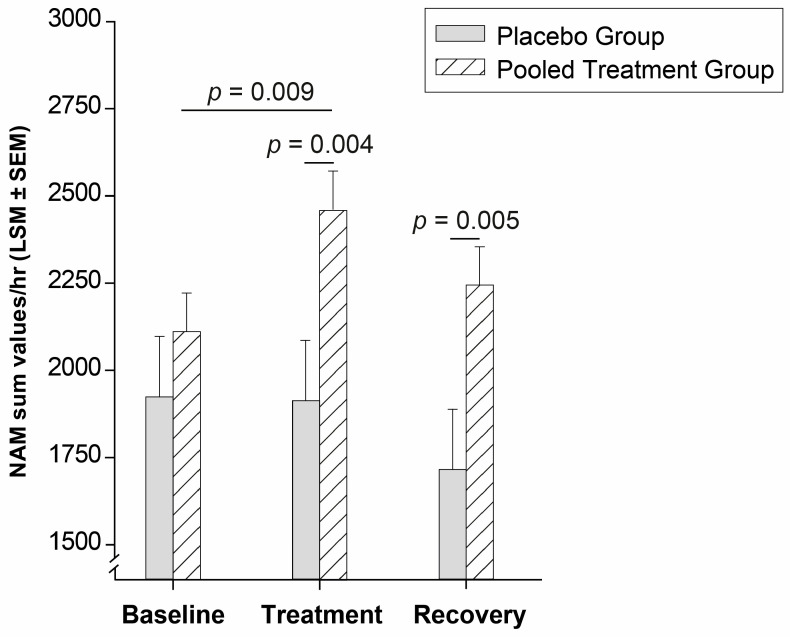
NAM sum values per hour evolution of both groups.

**Figure 3 animals-14-00711-f003:**
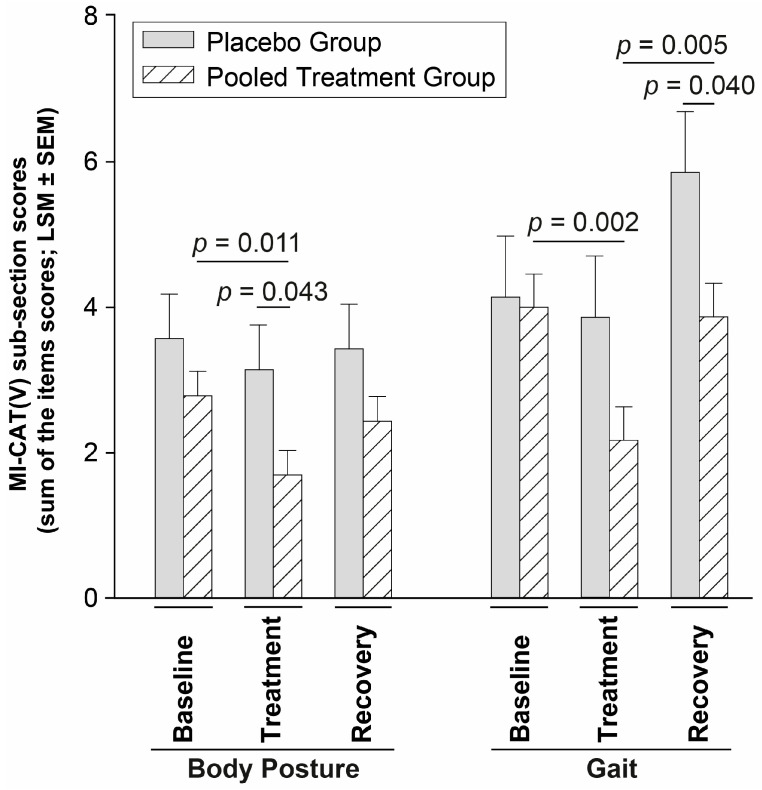
MI-CAT(V) evolution of the gait and body posture sub-sections.

**Figure 4 animals-14-00711-f004:**
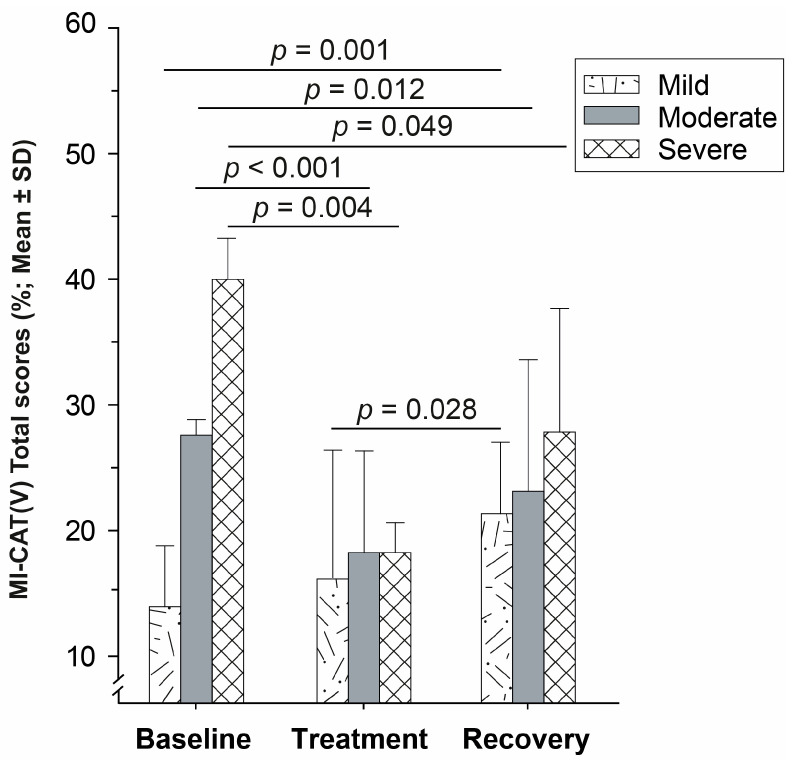
Within-time evolution of the MI-CAT(V) total score for each cluster.

**Table 1 animals-14-00711-t001:** Correlations between the MI-CAT(V) total score or sub-sections and functional assessments at Baseline.

	Path—Velocity	Path—Time of Completion	Path—Thoracic Limb PVF	NAM	Stairs—Finish Line Up
**MI-CAT(V)**	−0.303 (*p* = 0.019)		**−0.501** (*p* < 0.001)	**−0.453** (*p* < 0.001)	
**Body Posture**				−0.352 (*p* = 0.006)	−0.258 (*p* = 0.047)
**Gait**			**−0.373** (*p* = 0.004)	**−0.449** (*p* < 0.001)	
**Obstacles**	**−0.405** (*p* = 0.001)	**0.408** (*p* = 0.002)	**−0.524** (*p* < 0.001)	**−0.404** (*p* = 0.002)	
**Global Dist. Exam.**	−0.280 (*p* = 0.030)		**−0.441** (*p* < 0.001)	**−0.430** (*p* < 0.001)	

The Spearman rank test coefficient of correlation (Rho_S_, ρ) and the associated *p*-value are presented when a statistically significant correlations was found (*p* < 0.05). Bold text indicates a moderate correlation, and an absence of text indicates no-correlation.

**Table 2 animals-14-00711-t002:** Responder cats (%) during Treatment and Recovery for the MI-CAT(V) total score and during Treatment for NAM.

	MI-CAT(V)	MI-CAT(V)	NAM
	Treatment	Recovery	Treatment
Placebo Group	14 ^a^	14	14 ^a^
Pooled Treatment Group	65 ^b^	39	78 ^b^
Mild	50 *	0 ^a^	70
Moderate	67	78 ^b^	78
Severe	100	50 ^a,b^	75

Responder cats (%) per cluster were calculated for each outcome as follows: a decrease in MI-CAT(V) over 15% vs. BSL and a positive or null slope after NAM linear representation (from W-5 to W2 for the treated cats or W-5 to W5 for the P. Gr.). Statistical differences were determined by a Fisher test (*p* < 0.05; different letters indicate an inter-group difference at the same timepoint whereas an * indicates a within-time difference). At Recovery, the number of responders was significantly lower for the mild cluster compared to the moderate cluster (*p* < 0.001) and tended to be different to the severe cluster (*p* < 0.066). During Treatment, the number of responders was greatly higher in the pooled treatment group compared to P Gr. either for MI-CAT(V) (*p* = 0.031) or NAM (*p* = 0.004).

## Data Availability

Data are contained within the article and Appendix A.

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
