# Peer review of "Concurrent Validation of MI-CAT(V), a Clinical Metrology Instrument for Veterinarians Assessing Osteoarthritis Pain in Cats, through Testing for Firocoxib Analgesic Efficacy in a Prospective, Randomized, Controlled, and Blinded Study"

_animals, 2024, doi:10.3390/ani14050711_

Round 1

Reviewer 1 Report

Comments and Suggestions for Authors

Thank you for your manuscript regarding concurrent validation of the MI-CAT(V) and efficacy testing of firoxocib for osteoarthritis pain in cats. Easy to use and validated CMIs are of great importance in the identification and treatment monitoring for feline OA and I commend you for your work in this field. 

Simple summary  

Line 18-20: “...which is associated with financial repercussions influencing the disease management and its detection, first.” I don’t think this portion of the sentence is necessary, especially in the simple summary where it is best to keep your key points, well, simple. I think your point is made in the first half of this sentence by saying OA is incurable and requires life-long treatment, then transition into the next sentence where you introduce the need for a diagnostic tool. 

Line 21-28: While this is all well written, the way it is presented here becomes a bit too technical and is written more as a second abstract rather than a true simple summary which a non-scientist could easily understand. Consider editing to make more approachable to the general public. 

Line 116: I disagree with your classification of your study population as geriatric cats. The American Association of Feline Practitioners classify geriatric cats to be 15 years or older and senior cats to be 11-14 years.  

Line 132: Please provide additional information on the severity scale used for radiographic scoring (perhaps in a table with the characteristics used to determine the 0-5 score). 

Line 172: As there is significant variation in the dimensions of stairs, can you please provide the width of each step and the type of surface (e.g. carpet, wood, etc.)? 

Line 372-373: The current phrasing of this sentence is a bit awkward. I think it would be helpful to specify the sample size was low “in these studies” and reword the end of the sentence where it says “...there is high risk it to be overestimated.” Additionally, I think it is important to mention that some of these prevalence studies are looking at radiographic evidence of degenerative joint disease (both appendicular joints and spondylosis deformans) and not always features of clinical pain. 

Comments on the Quality of English Language

Line 32: “Cats naturally affecting by OA...” the word “affecting” here should be “affected” 

Line 39: “Cats into moderate and severe” the word “into” here should be “in” 

Line 41: “...seemed less responder...” this phrase should be edited to “responded to a lesser degree” or something similar. 

Line 70: “...and coat and claws change...” the wording should be edited to “...and coat and claw changes...” 

Line 124: Cats were weighted...” the word “weighted” here should be “weighed” 

Line 144: “...purported to relief or ease...” the word “relief” here should be “relieve” 

Line 235: “...a decreased in MI-CAT(V)...” the word “decreased” here should be “decrease” 

Line 282: “...overtime.” should be two words “...over time.” 

Line 373-374: ...presents the paradox to be highly prevalent, and in the same time...” requires editing, such as “...present the paradox of a high prevalence, while at the same time being...”

Author Response

  1. Reviewer 1

Thank you for your manuscript regarding concurrent validation of the MI-CAT(V) and efficacy testing of firoxocib for osteoarthritis pain in cats. Easy to use and validated CMIs are of great importance in the identification and treatment monitoring for feline OA and I commend you for your work in this field.

Author response: Thank you for the time you spent to give us feedback, we really appreciate it. We considered your comments carefully and we believe that this new version of the manuscript will meet your high standard.

Response to comment 1: Line 18-20: “...which is associated with financial repercussions influencing the disease management and its detection, first.” I don’t think this portion of the sentence is necessary, especially in the simple summary where it is best to keep your key points, well, simple. I think your point is made in the first half of this sentence by saying OA is incurable and requires life-long treatment, then transition into the next sentence where you introduce the need for a diagnostic tool.

Thank you for your comment. We agreed with your comment and modified the text, accordingly.

Response to comment 2: Line 21-28: While this is all well written, the way it is presented here becomes a bit too technical and is written more as a second abstract rather than a true simple summary which a non-scientist could easily understand. Consider editing to make more approachable to the general public.

We really appreciated your comment, the simple summary was revised, and we hope you find the revisions to be satisfactory (l21-24).

Response to comment 3: Line 116: I disagree with your classification of your study population as geriatric cats. The American Association of Feline Practitioners classify geriatric cats to be 15 years or older and senior cats to be 11-14 years. 

Thank you for pointing this out, we agreed with you and corrected the sentence by removing the term ‘geriatric’, to only keep ‘adult’ (l118). We also retired it at (l185) referring about Path outcome.

Response to comment 4: Line 132: Please provide additional information on the severity scale used for radiographic scoring (perhaps in a table with the characteristics used to determine the 0-5 score).

Thank you for your relevant comment. The radiographic score corresponds to the summation of the (Likert-type) severity scale of the twelve appendicular joints evaluated, added to the number of joints affected by radiographic OA.  A Diplomate of the American College of Veterinary Surgeons (B.LU.) reviewed and scored all X-rays independently and blindly. You will find below the table summarising the severity scale, we are not sure of the relevance of adding it to the manuscript since we described it in the text. We leave it to the editor to decide. However, in a previous article, we appended the radiographic scores of all the cats included in the study (Appendix A; Delsart et al., 2022; IJMS). We can add them in Appendix if you wish.

Table 1 – Severity scale used to determine the radiographic score.

Severity scale

Description

0

No radiographic abnormality

1

Early signs of OA such as presence of osteophytes and/or sclerosis and/or enthesiophytes* and/or mineralization*

2

Mild OA

3

Mild/Moderate OA

4

Moderate OA

5

Severe OA

*enthesiophytes and mineralization must be accompanied by other radiographic signs (e.g., osteophytes and sclerosis) to be considered as significant.

Response to comment 5: Line 172: As there is significant variation in the dimensions of stairs, can you please provide the width of each step and the type of surface (e.g. carpet, wood, etc.)?

Thank you for your relevant comment, all the cats were assessed in the same establishment and in the same stairs. It is made with ceramic tiles and one step dimension is: height of 20 cm, width of 117 cm and depth of 28 cm. These characteristics were added in the manuscript (l181-182). More details are presented in our previous publication referenced as [6] in the corresponding text.

Response to comment 6: Line 372-373: The current phrasing of this sentence is a bit awkward. I think it would be helpful to specify the sample size was low “in these studies” and reword the end of the sentence where it says “...there is high risk it to be overestimated.” Additionally, I think it is important to mention that some of these prevalence studies are looking at radiographic evidence of degenerative joint disease (both appendicular joints and spondylosis deformans) and not always features of clinical pain.

Thank you for your suggestion. We reworded the sentence and specified that studies were both looking at radiographic and clinical evidence (l389-391).

Response to the comments on the quality of English language:

We really appreciated all your comments, thank you for the time you spent to improve the quality of our manuscript, we taken all in consideration (l32, l41, l70, l124, l146, l244, l296, l393).

Reviewer 2 Report

Comments and Suggestions for Authors

I have read the manuscript very carefully and find the study very interesting. I have some doubts about the statistical analysis, which is probably correct, however the authors should specify what each of the statistical tests used was used for, also for agreement between observers it is necessary to specify which test was used. Furthermore, it is appropriate to evaluate the power of the sample or at least the power of the effect

Comments on the Quality of English Language

Minor editing of English language required

Author Response

2. Reviewer 2

I have read the manuscript very carefully and find the study very interesting. I have some doubts about the statistical analysis, which is probably correct, however the authors should specify what each of the statistical tests used was used for, also for agreement between observers it is necessary to specify which test was used. Furthermore, it is appropriate to evaluate the power of the sample or at least the power of the effect.

Authors response: Thank you for your precious feedback. As per your comments, we improved the statistical paragraph by adding the purpose of the tests and the power of the effect (l226, l292 for MI-CAT(V) power, l304 for NAM power). The power of the effect was 83% for the MI-CAT(V) and 99% for NAM for the intergroup comparison. We have not calculated the inter-observer agreement since this study was carried out with a single evaluator. We are aware of the importance of validating the scale between observers and this will be the subject of a future article to be published this year. The model used to determine the intraclass coefficient correlation was added (l220). We believe that this revised version of the manuscript will now meet you high standards.

Author response to comments on the quality of English language: Thank you for your comment. We modified some sentences and words in the text to improve the quality of the manuscript (in yellow).

Reviewer 3 Report

Comments and Suggestions for Authors

The article presents an appropriate design with clear criteria and objective questions. Good inclusion/exclusion criteria. Well-defined and clear groups, facilitating a temporal/quantitative/qualitative assessment of the overall data. Regarding statistical analysis, the authors should, as in Table 1, provide in the section the "p" values associated with the ICC. The correlation coefficient values should be linked to the "p" values obtained for each of the data sets. Specify the normality test adopted by the authors.

Author Response

3. Reviewer 3

The article presents an appropriate design with clear criteria and objective questions. Good inclusion/exclusion criteria. Well-defined and clear groups, facilitating a temporal/quantitative/qualitative assessment of the overall data. Regarding statistical analysis, the authors should, as in Table 1, provide in the section the "p" values associated with the ICC. The correlation coefficient values should be linked to the "p" values obtained for each of the data sets. Specify the normality test adopted by the authors.

Authors response: Thank you for the time you spent to revise the manuscript. We appreciate your comments, we specified the “p” values associated with the ICC (l258) and the normality test used (l231). We believe that this revised manuscript will meet your high standards.

Response to comment: Could the training period mask the effect of Firocoxib?

The training phase does not involve physical exercise, but rather getting the cat used to the procedure. This period is essential, as it helps to reduce the cat's stress during the assessment. In fact, stress could be an evaluation bias (stress-induced analgesia) that’s why the training is necessary. However, the time allocated to this training and its nature is not sufficient for there to be an effect of the exercise on OA pain. Following your comment, we verified the term “training” to not be used for “acclimation”, as it could indeed generate some confusion.

Round 2

Reviewer 2 Report

Comments and Suggestions for Authors

Dear authors, you have answered my questions exhaustively. The manuscript can be accepted in this form.

Comments on the Quality of English Language

 Minor editing of English language required